# Divide and Conquer: Exploring Language-centric Tree Reasoning for Video Question-Answering

Zhaohe Liao [* 1 4]   Jiangtong Li [* 2 3]   Siyu Sun [1]   Qingyang Liu [1]   Fengshun Xiao [4]   Tianjiao Li [4]   Qiang Zhang [4]
Guang Chen [3]   Li Niu [1]   Changjun Jiang [2 3]   Liqing Zhang [1]

## Abstract

Video Question-Answering (VideoQA) remains challenging in achieving advanced cognitive reasoning due to the uncontrollable and opaque reasoning processes in existing Multimodal Large Language Models (MLLMs). To address this issue, we propose a novel Language-centric Tree Reasoning (LTR) framework that targets on enhancing the reasoning ability of models. In detail, it recursively divides the original question into logically manageable parts and conquers them piece by piece, enhancing the reasoning capabilities and interpretability of existing MLLMs. Specifically, in the first stage, the LTR focuses on language to recursively generate a language-centric logical tree, which gradually breaks down the complex cognitive question into simple perceptual ones and plans the reasoning path through a RAG-based few-shot approach. In the second stage, with the aid of video content, the LTR performs bottom-up logical reasoning within the tree to derive the final answer along with the traceable reasoning path. Experiments across 11 VideoQA benchmarks demonstrate that our LTR framework significantly improves both accuracy and interpretability compared to state-of-the-art MLLMs. To our knowledge, this is the first work to implement a language-centric logical tree to guide MLLM reasoning in VideoQA, paving the way for language-centric video understanding from perception to cognition.

---

[*]Equal contribution , [1] MoE Key Lab of Artificial Intelligence, Shanghai Jiao Tong University, Shanghai China; [2] Key Laboratory of Embedded System and Service Computing, Ministry of Education, Shanghai, China; [3] School of Computer Science and Technology, Tongji University, Shanghai, China; [4] Bilibili Inc, Shanghai, China. Correspondence to: Liqing Zhang <lqzhang@cs.sjtu.edu.cn>.

*Proceedings of the 42nd International Conference on Machine Learning*, Vancouver, Canada. PMLR 267, 2025. Copyright 2025 by the author(s).

## 1. Introduction

Video Question-Answering (VideoQA) has emerged as a significant research area with applications in multi-modal understanding, interactive artificial intelligence and cognitive science (Xiao et al., 2021; Li et al., 2022a; Wu et al., 2023; Li et al., 2023b; Mangalam et al., 2023; Li et al., 2024b; Chen et al., 2022; Wang et al., 2025; Chi et al., 2024a; Li et al., 2025; Chi et al., 2024b). The core challenge lies in advancing language-centric video understanding from perception to cognition. Specifically, low-level perception involves understanding the spatiotemporal features of videos, such as recognizing objects, actions, and scenes. High-level cognition, on the other hand, requires comprehending the underlying logic of both the video content and the posed questions, enabling the system to perform reasoning along a logical structure and provide accurate answers. To address these challenges, recent advances extend Large Language Models (LLMs) to their multimodal variants (MLLMs), such as Video-LLaMA (Zhang et al., 2023; Cheng et al., 2024) and Video-LLaVA (Lin et al., 2024) by integrating visual and textual information.

However, while they can provide certain explanations when answering questions, how to achieve System-2 reasoning has not been fully explored. One major limitation is that the reasoning processes of these models are often uncontrollable and lack of transparency. Such opacity makes it challenging to analyze their reasoning steps, rendering the results to be less trustworthy. For instance, when addressing complex questions involving multiple temporal visual cues, the models may produce incorrect answers without revealing the reasoning path that led to those conclusions. Therefore, as users cannot trace back to pinpoint where the reasoning have gone astray, they cannot trust the results.

To overcome such issues, we propose a novel training-free, model-agnostic **Language-centric Tree Reasoning (LTR)** framework that enhances model reasoning capabilities while improving the interpretability and verifiability of reasoning processes. Our framework utilizes language as the central driving force for video understanding, starting from the logical structure inherent in the question itself. Initially, by integrating video content, we recursively generate a com-

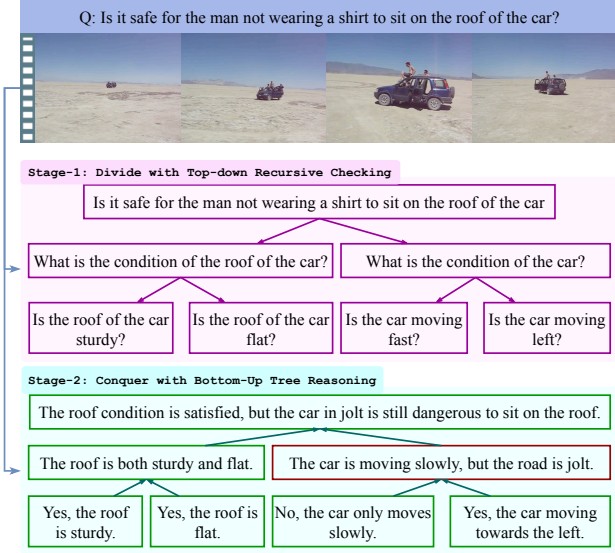

*Figure 1.* The example about the reasoning process of our LTR framework. In the first stage, LTR recursively divide the question into language-centric logical tree. In the second stage, LTR conquer the main question as bottom-up tree reasoning. Note that due to the assistance from video content, some flaw in logical tree can also be fixed during bottom-up tree reasoning.

prehensive *language-centric logical tree* from the given question. This tree has two signatures: first, its hierarchical structure explicitly represents the *reasoning logic* of the original question; second, the leaf nodes comprise simple *perceptual* questions that enable effective leveraging of existing MLLMs' perceptual strengths. To generate such a logical tree, the MLLM recursively divides the questions into simpler and logically reasonable sub-questions and decides whether the sub-questions are simple enough to be leaf perceptual questions. Subsequently, to perform multimodal System-2 reasoning with the generated language-centric logical tree, we begin by employing MLLMs to answer all leaf node questions, establishing an evidentiary foundation for the bottom-up reasoning process. Then, aided by video content, we perform logical bottom-up reasoning recursively within the tree, deriving node answers from the responses of their children while verifying consistency with visual evidence, ultimately obtaining both the original question's solution and a fully traceable reasoning path.

Some previous works have also explored interpretable VideoQA method, such as VoT (Fei et al., 2024a) and DSTN (Qian et al., 2022). VoT constructs a cognitive-level reasoning framework by providing detailed analysis at the object and action levels, followed by reasoning based on fine-grained video representations. While VoT enhances MLLM performance and offers additional reasoning cues, it still falls short in capturing the logical structure of questions and fully elucidating the reasoning process. Unlike VoT, our

reasoning framework is designed on the language-centric logical tree, improving the verifiability and facilitating further errors analysis. As another approach, DSTN exploits Neural Modular Networks (NMNs) to generate a program and obtains the final answer through program execution. Although this approach enhances verifiability, it lacks error tolerance, *i.e.*, any mistake in the program invariably leads to an unrecoverable incorrect answer. In contrast, our framework offers a soft reasoning architecture, supplementing logical reasoning with video information, ensuring explainable reasoning while increasing error tolerance.

To validate the effectiveness of our LTR framework, we select four existing MLLMs, *i.e.* VideoLLaMA3 (Zhang et al., 2025), VideoChat2 (Li et al., 2024b), Qwen2-VL (Wang et al., 2024), and LLaVA-OneVision (Li et al., 2024a) as baselines for comparison, and conduct experiments on 11 VideoQA benchmarks. Additionally, we perform ablation studies to analyze the effectiveness of each component within our framework and provide insights into the strengths and weaknesses of existing models based on our framework. Moreover, we provide case studies to demonstrate how our framework enables more error-tolerant, explainable, and controllable System-2 reasoning while enhancing prediction accuracy. Our contribution can be summarized as:

- **Motivation**: We analyze the two main development stages of VideoQA, *i.e.*, perception and cognition, to explore a language-centric tree reasoning framework that achieves cognitive-level video understanding.

- **Framework**: We propose a novel training-free, model-agnostic **Language-centric Tree Reasoning (LTR)** framework that utilizes language as the central driving force for video understanding, enhancing reasoning ability and interpretability of MLLMs.

- **Experiments**: Extensive experiments on 11 benchmarks against 4 baseline MLLMs demonstrate that the LTR framework improves reasoning accuracy, leading to a more transparent and verifiable VideoQA system.

## 2. Related Work

### 2.1. Video Question-Answering Dataset

By incorporating the temporal dimension, VideoQA naturally extends ImageQA, enabling the ability to answer questions with dynamic visual content. In recent years, VideoQA datasets have evolved to encompass more complex tasks that require advanced reasoning skills, such as temporal reasoning (Choi et al., 2021), physical reasoning (Yi et al., 2020), evidence reasoning (Xiao et al., 2021; Wu et al., 2023), commonsense reasoning (Li et al., 2022a; 2023b), and long video understanding (Mangalam et al., 2023). To

enhance fine-grained video understanding, AGQA (Grunde-McLaughlin et al., 2021) and AGQA-Decomp (Gandhi et al., 2022) utilize spatiotemporal scene graphs from Action Genome (Ji et al., 2020) to construct VideoQA datasets. With the recent advancements of MLLMs in video understanding, test-only benchmarks such as MVBench (Li et al., 2024b), MMT-Bench (Ying et al., 2024), and Video-MME (Fu et al., 2024) have been proposed to comprehensively evaluate MLLM performance. However, the reasoning logic in answer prediction remains under-explored. Our framework leverages the zero-shot reasoning ability of MLLMs to generate high-quality fine-grained logical trees when answering questions, reliefing the constraint caused by the absence of such datasets.

## 2.2. Video Question-Answering Methodology

Prior to the incorporation of Large Language Models (LLMs) in VideoQA, the architectures of VideoQA methods primarily focused on aligning videos with questions. In early studies, this video-question alignment is achieved through the use of cross-modal attention (Li et al., 2019; Gao et al., 2019) or memory networks (Gao et al., 2018; Fan et al., 2019). Subsequently, graph reasoning (Jiang & Han, 2020; Park et al., 2021; Liu et al., 2021; Gu et al., 2021; Cherian et al., 2022) and hierarchical reasoning (Le et al., 2020; Guo et al., 2021; Xiao et al., 2022a; Peng et al., 2022; Xiao et al., 2022b) approaches gained popularity. Some research has explored VideoQA from a causality perspective. For example, IGV (Li et al., 2022c), EIGV (Li et al., 2022b), and TIGV (Li et al., 2024d) focus on distinguishing causal and environmental clips in video by utilizing a simple grounding indicator and promoting sensitivity to semantic changes in causal scenes. Furthermore, KPI (Li et al., 2023a) investigates front-door interventions as knowledge proxy to mitigate the effects of dataset biases.

With the advancement of Large Language Models (LLMs), Video-MLLMs generally comprise a pre-trained visual encoder (Radford et al., 2021; Caron et al., 2021; Fei et al., 2024b) that encodes video frames into low-dimensional visual features, a vision-language aligner to ensure that these visual representations are comprehensible to LLMs, and a language decoder instruction-tuned on multimodal data (Cai et al., 2024; Dubey et al., 2024) that generates text responses based on provided instructions and video content. Numerous efforts have been dedicated to designing effective visual alignment strategies. For example, Video-ChatGPT (Maaz et al., 2023) employs spatial and temporal pooling to independently caption information. VideoLLaMA2 (Cheng et al., 2024) presents the Spatial-Temporal Convolution Connector to more efficiently capture spatial-temporal features with a manageable number of visual tokens. Furthermore, Video-LLaVA (Lin et al., 2024) aligns video and image data prior to projection, achieving unified visual representation

and better computational efficiency.

Despite these advancements, existing MLLMs still provide limited explanations for their answering process. To address these challenges, we propose the LTR framework which explores language-centric logical tree reasoning in VideoQA, thereby enhancing the reasoning transparency.

## 2.3. Visual Reasoning

In the field of visual reasoning, the effectiveness of dividing complex questions into simpler sub-questions has been observed across various tasks, including ImageQA (Cao et al., 2018) and DocumentQA (Yang et al., 2018). In earlier research, most benchmarks (Hudson & Manning, 2019; Grunde-McLaughlin et al., 2021) break down questions into modular programs defined within Neural Modular Networks (NMNs) (Qian et al., 2022) to generate answers. Among these, AGQA (Grunde-McLaughlin et al., 2021) introduces spatio-temporal scene graphs to represent reasoning programs for VideoQA. However, these reasoning programs from AGQA (Grunde-McLaughlin et al., 2021) cannot be directly utilized by existing VideoQA methods. To address this limitation, AGQA-Decomp (Gandhi et al., 2022) transforms each reasoning program into multiple sub-questions and a compositional graph to evaluate the compositional consistency of VideoQA methods. Neural Modular Networks (NMNs) have been extensively explored in both ImageQA (Andreas et al., 2016; Hu et al., 2017; Johnson et al., 2017; Mascharka et al., 2018) and VideoQA (Qian et al., 2022) for performing compositional reasoning in visual question answering. Chen et al. (2024) introduces general and universial meta-points for object represent, which effectively facilitate the reasoning on object interaction. These networks convert questions into modular sub-programs and assemble these modules to execute the overall reasoning process. Although NMN-based approaches offer interpretability, they rely on predefined modules, limiting the compatibility with novel modules and reasoning frameworks.

In the era of MLLMs, VoT (Fei et al., 2024a) endeavors to establish transparent VideoQA reasoning by offering detailed analyses of videos at the object and action levels and conducting reasoning based on fine-grained video representations. However, it still fails to capture the logical structure of questions or fully elucidate the reasoning process. Logic-CheckGPT (Wu et al., 2024) addresses object hallucination and enhances visual understanding ability of MLLMs by checking conflict consistency across multiple interrelated questions. However, this method neither explores hierarchical question structures spanning cognitive reasoning to perceptual processing nor provides traceable reasoning paths. Therefore, we introduce the LTR framework, which first recursively generates a language-centric logical tree and then performs visual-assisted reasoning along this logical tree. In

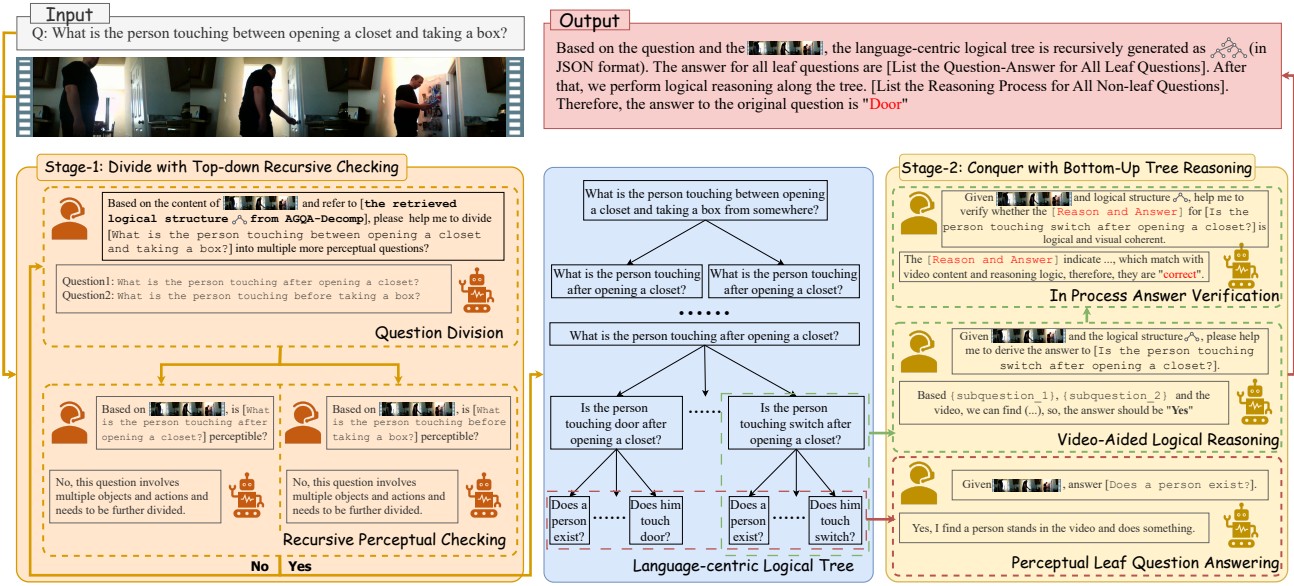

Figure 2. The illustration of Language-centric Tree Reasoning (LTR) framework. In the first stage, our LTR recursively divides complex cognitive questions to simpler question until they are perceptual questions. In the second stage, our LTR answers the perceptual leaf questions and bottom-up reasoning toward the original question alone the language-centric logical tree. Best viewed when zoomed in.

this way, LTR method enhances both reasoning capabilities and interpretability, advancing towards System-2 reasoning.

## 3. LTR Framework

### 3.1. Divide with Top-down Recursive Checking

In this stage, our LTR recursively breaks down the cognitive question into perceptual ones to construct the language-centric logical tree with two steps. The first step divides non-perceptual questions into sub-questions, and the second step recursively checks whether the sub-questions are perceptual to determine the stop of question division.

#### 3.1.1. QUESTION DIVISION

The Question Division step is motivated by the need to progressively divide complex cognitive questions into simpler perceptual ones. However, directly dividing such questions based solely on textual content proves challenging due to their inherent reliance on comprehensive video understanding. To address this difficulty, this step leverages video To address this, the step strategically utilizes visual evidence to guide hierarchical division. Specifically, given a question and its corresponding video input, the MLLM recursively divides the original question into logically structured sub-questions that facilitate stepwise reasoning. Through this recursive process, LTR reduces cognitive complexity by transforming abstract queries into concrete perceptual tasks.

However, directly asking the model to perform question division may introduce uncertainty, such as multiple vi-

able division approaches and ambiguity in determining perceptual simplicity. Therefore, we introduce retrieval-augmented generation (RAG)-assisted question division. In detail, we use the AGQA-Decomp (Gandhi et al., 2022) dataset as the retrieval database and embed all non-leaf question-subquestion tuple in it. Specifically, for each non-leaf question $q \in Q$ (where $Q$ denotes the complete non-leaf question set), we encode it into a 256-dimensional feature vector $f_q \in \mathbb{R}^{256}$ using OpenAI's `text-embedding-3-large` model, with this vector serving as the retrieval key. During the RAG process, we retrieve the top-$K$ most relevant question-subquestion tuples from AGQA-Decomp by computing cosine similarity between feature vectors. These retrieved tuples then provide few-shot exemplars for subquestion generation. To facilitate this process, we prompt the MLLM with:

---

**Instruction**: Based on the content of the given video, please help me to divide the given question into multiple more perceptual questions.
**Sample**: The similar samples for the question division are: {retrieval_results_in_JSON}.
**Rule**: The output should be formed in JSON format, with $SubQ_i$: {sub-question} as the key and value.
**Question**: {original_question}.
**Video:** {video_tokens}

---

#### 3.1.2. RECURSIVE PERCEPTUAL CHECKING

The Recursive Perceptual Checking step is used to determine whether a given question can be directly answered

with the perceptual capabilities of MLLMs. However, the feasibility of directly answering a certain question may vary as the video content changes. For example, the question "Is it safe to drive ahead?" can be directly answered if the video depicts an empty road, as the assessment of safety is straightforward. In contrast, if the video contains a complex urban environment with multiple obstacles and dynamic elements, we need to divide the question into several sub-questions focusing on specific objects and actions. Each sub-question would then be individually addressed to collectively form a comprehensive answer. Therefore, we input both the question and the video into the MLLM. The MLLM analyzes the combined inputs to judge whether, for the video, the question qualifies as a perceptual query that can be answered directly. The perceptual questions will not be further divided. When all leaf questions are determined as perceptual, the top-down recursive checking stops, and the language-centric logical tree is generated accordingly. To facilitate this process, we prompt the MLLM with:

> **Instruction**: Based on the given video, is the given question perceptible to the video?
> **Rule**: The output should be "Yes/No + *Reason*"
> **Question**: {original_question}.
> **Video:** {video_tokens}

### 3.2. Conquer with Bottom-up Tree Reasoning

In the second stage, LTR performs bottom-up tree reasoning through three steps to recursively derive a logical and visual coherent answer. In the first step, LTR answers all perceptual leaf questions based on video content. In the second and third steps, aided by video content, LTR reasons along the first-order logical structures and verifies answers, ensuring visual and logical coherent of answers.

#### 3.2.1. PERCEPTUAL QUESTION ANSWERING

The Perceptual Question Answering step aims to address the leaf questions of the language-centric logical tree, thereby enabling further reasoning along the tree structure. Note that these leaf questions are simple enough to be perceptible from the video, as depicted in Section 3.1.2. Therefore, they can be effectively solved by leveraging the strong perceptual capabilities of existing MLLMs. In this step, we adopt the conventional instruction used in MLLMs for VideoQA. Specifically, we input the question and the video into the MLLM to generate the corresponding answer. To facilitate this operation, we prompt the MLLM with:

> **Instruction**: Given the question and video, answer the question using several words or phrase.
> **Question**: {original_question}.
> **Video:** {video_tokens}

#### 3.2.2. VIDEO-AIDED LOGICAL REASONING

The Video-aided Logical Reasoning step is designed to infer the answer to a parent question within the language-centric logical tree by utilizing the answers to its sub-questions, with the assistance of video information. The motivation behind this step stems from recognizing that potential vague may occur during the question decomposition process, and the answers to sub-questions may also contain inaccuracies. Therefore, relying solely on linguistic information for logical reasoning might not yield reliable results. By incorporating video content, we enhance the reasoning process, allowing the MLLM to consider visual cues that may not have been explicitly addressed in the sub-questions. For instance, as illustrated in Figure 1, even though the sub-questions do not include inquiries about the stability of the road, the MLLM still accounts for road bumps when answering about the "condition of the car". In practice, we input the video, the parent question, and the sub-questions along with their answers into MLLM to perform video-aided logical reasoning. This approach ensures that the reasoning is grounded in both logic and video, mitigating the impact of potential errors in logical tree or sub-question answers. To facilitate this process, we prompt the MLLM with:

> **Instruction**: Given the video and the logical structure, please help me to derive the answer to the main question.
> **Rule**: The output should first provide explanation from both logic and video, and then provide the final answer.
> **Logical Structure**: {logical_structure_in_JSON_format}.
> **Video:** {video_tokens}

#### 3.2.3. IN-PROCESS ANSWER VERIFICATION

The In-process Answer Verification step is designed to ensure the reliability of intermediate answers during the reasoning process, which involves logical reasoning across multiple intermediate nodes. Before progressing further, it is essential to verify these intermediate answers to reduce the propagation of errors. Specifically, our verification focuses on two key aspects: (1) whether the answers and reasoning processes of intermediate node questions conform to the logical reasoning from sub-questions to parent questions, and (2) whether the answers to intermediate node questions conflict with the video content itself. The first aspect emphasizes cognitive-level verification, assessing the logical consistency and coherence of the reasoning process. The second aspect concentrates on perceptual-level verification, evaluating the answers against the basic perceptual information derived from the video content. By integrating both cognitive and perceptual validations, we effectively ensure the accuracy and rationality of the final answers. Notably, when this step detects any cognitive or perceptual conflicts in the intermediate node answers, we

incorporate the conflicts as inputs and re-execute the Video-aided Logical Reasoning step to re-solve them. To facilitate this verification process, we prompt the MLLM with:

**Instruction**: Given the video and the logical structure, help me to verify whether the [*Reason and Answer*] for the main question is logical and visual coherent.
**Rule**: The output should first evaluate the logical and visual coherency, and then judge the correctness.
**Logical Structure**: {logical_structure_in_JSON_format}.
**Video:** {video_tokens}

## 4. Experiment

### 4.1. Evaluation Protocols

We evaluate the LTR framework on 11 VideoQA benchmarks, including MSVD-QA (Xu et al., 2016), MSRVTT-QA (Xu et al., 2016), TGIF-QA (Jang et al., 2017), ActivityNet-QA (Yu et al., 2019), AGQA-Decomp (Gandhi et al., 2022), NExT-QA (Xiao et al., 2021), Causal-VidQA (Li et al., 2022a), STAR (Wu et al., 2023), Ego-Schema (Mangalam et al., 2023), Video-MME (Fu et al., 2024), and MVBench (Li et al., 2024b). For open-ended settings, we use GPT-3.5 to assess the generated responses, utilizing both accuracy and scoring metrics. For multiple-choice settings, we use the corresponding MLLM to select answers from the provided options based solely on the question and generated response. To illustrate improvements in compositional consistency, we evaluate compositional metrics (cR, cP, c-$F_1$) from VA³ (Liao et al., 2024) on AGQA-Decomp. **Full experiment results are in the Appendix.**

### 4.2. Implementation Details

In our LTR framework, we utilize four different MLLMs: VideoLLaMA3 (Zhang et al., 2025), VideoChat2 (Li et al., 2024b), Qwen2-VL (Wang et al., 2024), and LLaVA-OneVision (Li et al., 2024a). We set the video resolution to $336 \times 336$ pixels and uniformly sample 16 frames from each video. The maximun new generated length is restricted to 2048 tokens. Other settings follow the recommended settings of zero-shot generations for each baseline model.

### 4.3. Comparisons on AGQA-Decomp

In Table 1, we compare the performance of LTR with 9 baseline methods on AGQA-Decomp. The columns marked with "main" and "sub" represents corresponding metrics computed on root question and none-root questions in the language-centric trees respectively. The experimental results indicate that our framework significantly outperforms the baselines in terms of accuracy, score, and compositional consistency, which can be attributed to the collaborative compositional reasoning strategy. To evaluate compositional

consistency, we utilize the DAG from the AGQA-Decomp test set for bottom-up tree reasoning. Regarding accuracy improvement, we observe more pronounced gains in sub-question compared to main questions. This is attributed to the relative simplicity of sub-questions, which facilitates more effective reasoning. Furthermore, the improvement in c$F_1$ is much larger than that in accuracy. This improvement is attributed to the *Video-aided Logical Reasoning* module, which exploits the logical relationships within the structure, enabling the QA inforamtion in perceptual questions to propagate along the tree and help the model answer more cognitive questions, therefore enhances the compositional consistency between main and sub questions.

### 4.4. Comparisons on Zero-Shot Performance

In Tables 2 to 4, we present zero-shot performance comparisons on three benchmarks: Causal-VidQA, NeXT-QA and MVBench, **with additional comparisons on other benchmarks provided in the Appendix**. Overall, our framework significantly outperforms the baselines, which is attributed to the collaborative *Divide with Top-down Recursive Checking* and *Conquer with Bottom-up Tree Reasoning* stages.

When comparing performance between simple perceptual grounding tasks and complex cognitive reasoning tasks, we find that LTR demonstrates larger improvements on relatively complex cognitive reasoning tasks. For example, Table 3 shows more significant improvements on counterfactual and prediction tasks (between 2.4% and 4.2%) than on explanation and description tasks (between 0.9% and 2.1%), as the former require more complex logical reasoning capabilities that are systematically facilitated by our language-centric tree reasoning procedure.

Specifically, the *Divide with Top-down Recursive Checking* stage guides MLLMs to extract necessary perceptual information for complex reasoning, while the *Conquer with Bottom-Up Tree Reasoning* stage gradually aggregates perceived visual clues through recursive logical reasoning to deduce answers through step-by-step complex reasoning. The combination of these two stages (*i.e.*, LTR framework) therefore enhances complex reasoning capabilities of MLLMs while maintaining traceable reasoning processes.

Similar patterns are also observed in MVBench (Table 2) and NeXT-QA (Table 4). Specifically, Table 2 reveals that improvements on reasoning-intensive tasks (*i.e.*, counterfactual inference (CI), episodic reasoning (ER), and action prediction (AP)) are more substantial than those on simple perceptual tasks (*i.e.*, object existence (OE), action count (AC), and fine-grained pose (FP)). Furthermore, Table 4 demonstrates more significant improvements on causal and temporal questions compared to descriptive questions.

These observations collectively confirm that our LTR frame-

| Method | Acc. | | Score | | Compositional Consistency | | |
|---|---|---|---|---|---|---|---|
| | main | sub | main | sub | cP | cR | cF$_1$ |
| Video-LLaVA (Zhang et al., 2023) | 57.5 | 65.0 | 2.7 | 3.3 | 58.7 | 59.6 | 59.1 |
| LLaMA-VID (Li et al., 2024c) | 58.2 | 63.2 | 2.8 | 3.2 | 56.1 | 58.8 | 57.4 |
| Chat-UniVi (Jin et al., 2024a) | 60.3 | 68.1 | 3.0 | 3.4 | 63.6 | 69.9 | 66.6 |
| VideoChat (Li et al., 2023c) | 56.4 | 63.7 | 2.7 | 3.3 | 57.4 | 56.7 | 57.1 |
| MiniGPT4-Video (Ataallah et al., 2024) | 54.5 | 61.8 | 2.4 | 2.1 | 55.6 | 52.8 | 54.1 |
| VideoChat2 (Li et al., 2024b) | 60.5 | 69.2 | 3.1 | 3.5 | 67.5 | 66.5 | 67.0 |
| + LTR | 64.3+3.8 | 75.3+6.1 | 3.4+0.3 | 4.1+0.5 | 77.2+9.7 | 77.7+11.2 | 77.4+10.4 |
| VideoLLaMA3 (Zhang et al., 2025) | 69.2 | 76.7 | 3.7 | 4.1 | 72.2 | 69.7 | 69.4 |
| + LTR | 72.7+3.5 | 82.7+6.0 | 4.0+0.3 | 4.5+0.4 | 80.4+8.2 | 82.1+12.4 | 81.2+11.8 |
| LLaVA-OneVision (Li et al., 2024a) | 64.3 | 73.2 | 3.5 | 3.9 | 69.1 | 66.9 | 68.0 |
| + LTR | 67.8+3.5 | 79.5+6.3 | 3.7+0.2 | 4.3+0.4 | 78.3+9.2 | 80.6+13.7 | 79.4+11.4 |
| Qwen2-VL (Wang et al., 2024) | 65.4 | 73.8 | 3.5 | 4.0 | 68.2 | 66.2 | 67.2 |
| + LTR | 69.1+3.7 | 80.7+6.9 | 3.8+0.3 | 4.4+0.4 | 78.6+10.4 | 80.7+14.5 | 79.6+12.4 |

*Table 1.* Performance on AGQA-Decomp regarding the accuracy, score, and compositional consistency. The columns marked with "main" and "sub" represents corresponding metrics computed on root question and none-root questions in the language-centric trees respectively. The results in the blue area are reproduced by us using their published model weights and instructions.

| Method | Avg | AS | AP | AA | FA | UA | OE | OI | OS | MD | AL | ST | AC | MC | MA | SC | FP | CO | EN | ER | CI |
|---|---|---|---|---|---|---|---|---|---|---|---|---|---|---|---|---|---|---|---|---|---|
| Video-LLaVA (Zhang et al., 2023) | 41.0 | 46.0 | 42.5 | 56.5 | 39.0 | 53.5 | 53.0 | 48.0 | 41.0 | 29.0 | 31.5 | 82.5 | 45.0 | 26.0 | 53.0 | 41.5 | 33.5 | 41.5 | 27.5 | 38.5 | 31.5 |
| LLaMA-VID (Li et al., 2024c) | 41.3 | 45.5 | 40.5 | 58.0 | 39.5 | 55.0 | 53.5 | 40.0 | 35.5 | 18.5 | 27.5 | 87.0 | 41.5 | 23.0 | 45.5 | 41.0 | 27.0 | 40.0 | 34.5 | 41.5 | 31.5 |
| LLaMA-Adapter (Zhang et al., 2024a) | 31.7 | 23.0 | 28.0 | 51.0 | 30.0 | 33.0 | 53.5 | 32.5 | 33.5 | 25.5 | 21.5 | 30.5 | 29.0 | 22.5 | 41.5 | 39.5 | 25.0 | 31.5 | 22.5 | 28.0 | 32.0 |
| Video-ChatGPT (Maaz et al., 2023) | 32.7 | 23.5 | 26.0 | 62.0 | 22.5 | 26.5 | 54.0 | 28.0 | 40.0 | 23.0 | 20.0 | 31.0 | 30.5 | 25.5 | 39.5 | 48.5 | 29.0 | 33.0 | 29.5 | 26.0 | 35.5 |
| VideoChat (Li et al., 2023c) | 35.5 | 33.5 | 26.5 | 56.0 | 33.5 | 40.5 | 53.0 | 40.5 | 30.0 | 25.5 | 27.0 | 48.5 | 35.0 | 20.5 | 42.5 | 46.0 | 26.5 | 41.0 | 23.5 | 23.5 | 36.0 |
| VideoLLaMA (Zhang et al., 2023) | 34.1 | 27.5 | 25.5 | 51.0 | 29.0 | 39.0 | 48.0 | 40.5 | 38.0 | 22.5 | 22.5 | 43.0 | 34.0 | 22.5 | 32.5 | 45.5 | 32.5 | 40.0 | 30.0 | 21.0 | 37.0 |
| VideoChat2 (Li et al., 2024b) | 60.3 | 66.5 | 74.0 | 85.5 | 51.0 | 61.5 | 85.5 | 68.0 | 43.5 | 48.5 | 35.5 | 83.5 | 38.5 | 66.5 | 88.0 | 50.5 | 63.5 | 46.5 | 36.0 | 42.5 | 70.5 |
| +LTR | 62.4 | 69.5 | 78.0 | 86.0 | 52.5 | 62.0 | 85.5 | 73.0 | 45.0 | 49.5 | 35.5 | 86.0 | 40.0 | 68.5 | 87.0 | 55.0 | 64.5 | 47.0 | 38.0 | 49.5 | 75.0 |
| VideoLLaMA3 (Zhang et al., 2025) | 67.1 | 70.5 | 72.5 | 91.5 | 43.5 | 85.5 | 92.5 | 74.5 | 42.0 | 51.5 | 44.5 | 92.5 | 53.0 | 75.0 | 92.0 | 59.0 | 61.5 | 76.5 | 33.5 | 54.0 | 75.5 |
| +LTR | 69.0 | 74.0 | 77.5 | 91.0 | 45.0 | 85.0 | 91.5 | 79.0 | 43.0 | 52.0 | 45.0 | 94.0 | 54.0 | 77.5 | 92.5 | 63.0 | 62.0 | 77.0 | 35.5 | 60.5 | 81.5 |
| LLaVA-OneVision (Li et al., 2024a) | 57.3 | 73.0 | 69.5 | 79.0 | 46.0 | 79.5 | 63.5 | 76.5 | 37.5 | 21.5 | 40.0 | 92.0 | 47.0 | 46.0 | 68.5 | 52.0 | 55.0 | 63.5 | 34.5 | 51.5 | 50.0 |
| +LTR | 59.9 | 79.5 | 76.5 | 79.0 | 46.5 | 80.0 | 62.0 | 82.0 | 38.5 | 23.5 | 41.0 | 91.0 | 48.0 | 48.0 | 69.5 | 57.5 | 56.0 | 65.0 | 37.0 | 59.5 | 57.5 |
| Qwen2-VL (Wang et al., 2024) | 65.7 | 77.0 | 80.5 | 81.5 | 49.5 | 75.0 | 93.5 | 72.5 | 39.0 | 47.0 | 47.5 | 93.5 | 46.0 | 81.5 | 94.5 | 46.5 | 58.5 | 68.0 | 41.5 | 55.0 | 66.5 |
| +LTR | 67.8 | 82.5 | 86.5 | 82.0 | 50.0 | 75.5 | 92.0 | 77.5 | 39.5 | 46.5 | 46.5 | 94.0 | 48.0 | 83.0 | 93.5 | 51.0 | 60.0 | 69.0 | 44.0 | 62.0 | 73.5 |

*Table 2.* Experimental results on MVBench. The results in the white area are copied from the corresponding works or MVBench (Li et al., 2024b), and the results in the blue area are reproduced by us using their published model weights and instructions.

work enhances cognitive reasoning capabilities of MLLMs more substantially than their perceptual abilities.

### 4.5. Ablation Study

In this section, we study the effect of the two reasoning stages in Table 5. The experiments are conducted on the Causal-VidQA and NeXT-QA with Qwen2-VL as baseline.

In the *Logical Tree Generation Stage*, we analyze the synergistic impacts of integrating RAG (line w/o RAG) and video content (line w/o video) during hierarchical tree construction. Experimental results indicate that incorporating RAG provides essential gudiance on the quality of generated language-centric trees. Moreover, the existence of video also provides spatiotemporal grounding cues within the LTR framework; without it, the tree lacks video-specific logical structures, leading to degraded performance.

In the *Logical Tree Reasoning Stage* stage, we examine the effects of answer verification (line w/o A.V.), correct perceptual answers (line w/o P.L.Q.A) and video incorporation (line R w/o video) in logical reasoning. Experimental results show that answer verification significantly impacts complex reasoning tasks by effectively identifying logical errors, while video incorporation is crucial since leaf questions alone cannot provide sufficient information for tree reasoning. Moreover, the results indicate that relatively precise perception is necessary for LTR framework. Although video-aided reasoning can provide some missing information during the process, it may not always be sufficiently accurate for deducing correct main answers. These findings highlight the importance of both answer verification and video incorporation in logical reasoning stage within the LTR framework, particularly for complex reasoning tasks.

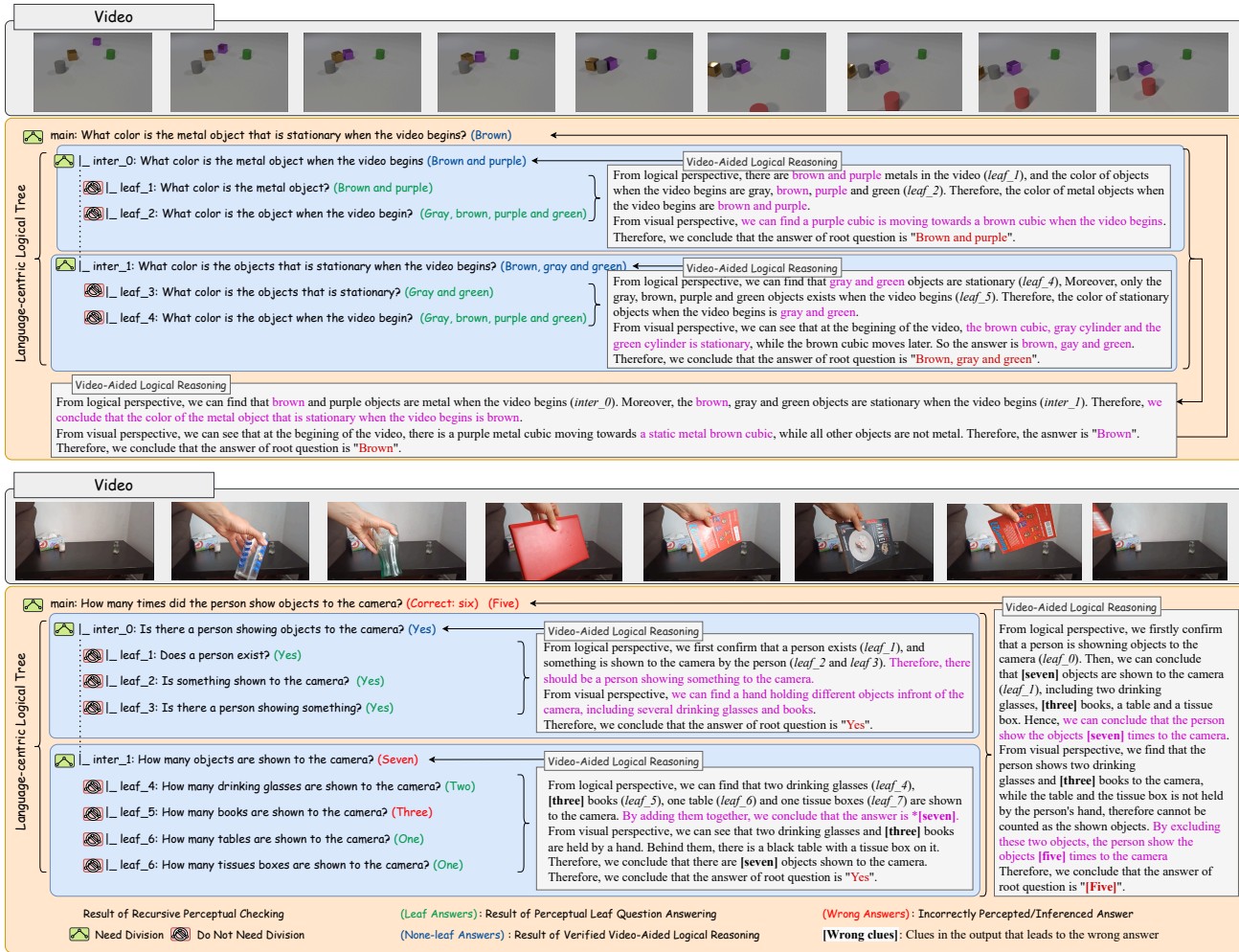

*Figure 3.* Qualitative examples on MVBench (Li et al., 2024b) generated by VideoChat2 (Li et al., 2024b) with our LTR framework.

## 5. Qualitative Results

In this section, we provide two extra qualitative results in Figure 3. The first case shows successful reasoning that uses perception results to derive the answer to the original question. The second case presents a failure scenario, showcasing that although our visual-aided logical reasoning can correct some errors in reasoning, severe perception incorrectness can also lead to unrecoverable failure.

Specifically, in the first video, a metal purple cube collides with a metal brown cube at the beginning, while all other objects are not metal. For sub-question *inter_0*, we further divide it to inquire about the colors from different perspectives. Based on the answers of *leaf_1* and *leaf_2*, our framework deduces the answer by finding the intersection of these two color sets, resulting in the correct answer "brown and purple". The sub-question *inter_1* is also processed in a similar manner. However, an object that is not stationary

in video could possibly be stationary at the beginning, as it may move in other parts of the video. Thanks to the visual-aided logical reasoning step, our framework successfully avoids this logical trap by recognizing that the brown cube is stationary at the beginning and starts moving later. Finally, for the main question, we conclude that the brown metal cube is stationary when the video begins.

In the second video, a man shows drink glasses to the camera twice and shows books to the camera four times sequentially. In this example, the sub-question *inter_0* is further divided into three perceptual questions, confirming the existence of the person, the object, and the action "show". Therefore, we conclude that the person does show something to the camera. Besides, the counting sub-question *inter_1* is divided into counting the different objects appearing in the video. The logic trap here is that some perceived objects serve as backgrounds and are not shown to the camera by the man. Therefore, the answer shall not be calculated by

| Method | Acc@D | Acc@E | Acc@P | Acc@C | Acc@A |
|---|---|---|---|---|---|
| Video-LLaVA (Zhang et al., 2023) | 73.7 | 74.4 | 47.7 | 51.5 | 61.8 |
| Video-ChatGPT (Maaz et al., 2023) | 73.1 | 75.1 | 46.0 | 50.0 | 61.1 |
| VideoChat (Li et al., 2023c) | 72.9 | 73.9 | 45.9 | 45.8 | 59.6 |
| VideoLLaMA (Zhang et al., 2023) | 69.2 | 71.0 | 44.4 | 45.0 | 57.4 |
| VoT (Fei et al., 2024a) | 81.2 | 83.0 | 54.7 | 58.6 | 69.4 |
| VideoChat2 (Li et al., 2024b) | 66.8 | 75.1 | 45.8 | 38.6 | 56.6 |
| + LTR | 67.8 | 77.2 | 48.8 | 42.8 | 59.1 |
| VideoLLaMA3 (Zhang et al., 2025) | 79.1 | 79.9 | 55.9 | 43.8 | 64.6 |
| + LTR | 80.0 | 80.8 | 58.3 | 47.5 | 66.6 |
| LLaVA-OneVision (Li et al., 2024a) | 78.6 | 78.0 | 53.1 | 44.2 | 63.5 |
| + LTR | 79.8 | 79.2 | 56.1 | 48.5 | 65.9 |
| Qwen2-VL (Wang et al., 2024) | 80.3 | 81.5 | 59.8 | 50.3 | 68.0 |
| + LTR | 81.4 | 82.3 | 62.3 | 54.8 | 70.2 |

*Table 3.* Zero-shot performance on Causal-VidQA. D: description, E: explanation, P: prediction, C: counterfactual, A: all. Acc@E and Acc@C are reported in answer and reason setting. The results in the blue area are reproduced by us using their published model weights and instructions.

| Method | Acc@D | Acc@T | Acc@C | Acc@All |
|---|---|---|---|---|
| Video-LLaVA (Zhang et al., 2023) | 75.9 | 63.8 | 67.7 | 66.3 |
| LLaMA-VID (Li et al., 2024c) | - | - | - | - |
| Video-ChatGPT (Maaz et al., 2023) | 75.7 | 64.1 | 66.9 | 64.4 |
| Video-LaVIT (Jin et al., 2024b) | - | - | - | - |
| VideoChat (Li et al., 2023c) | 74.6 | 61.5 | 63.5 | 61.8 |
| VideoLLaMA (Zhang et al., 2023) | 72.3 | 57.4 | 59.2 | 60.6 |
| VoT (Fei et al., 2024a) | 83.3 | 74.6 | 75.8 | 76.0 |
| VideoChat2 (Li et al., 2024b) | 84.6 | 76.0 | 78.6 | 78.8 |
| + LTR | 85.8 | 78.7 | 81.6 | 81.4 |
| VideoLLaMA3 (Zhang et al., 2025) | 87.7 | 81.0 | 82.7 | 83.0 |
| + LTR | 88.5 | 82.9 | 84.8 | 84.8 |
| LLaVA-OneVision (Li et al., 2024a) | 85.5 | 76.3 | 78.6 | 79.0 |
| + LTR | 86.4 | 78.9 | 82.2 | 81.8 |
| Qwen2-VL (Wang et al., 2024) | 85.7 | 78.2 | 80.1 | 80.4 |
| + LTR | 86.7 | 80.3 | 82.8 | 82.7 |

*Table 4.* Experimental results on NExT-QA. D: descriptive, T: temporal, C: causal. The results in the white area are copied from the corresponding works or VoT (Fei et al., 2024a), and the results in the blue area are reproduced by us using their published model weights and instructions.

simply adding the numbers of sub-question answers. Moreover, the MLLMs falsely count the number of books shown to the camera as three. These two challenges lead to an incorrect answer of "seven" for this sub-question. However, in the higher-level logical reasoning (*i.e.*, the visual-aided logical reasoning of the main question), the logic trap is partially avoided by the visual assistance since our framework finds that the table and tissue box are not interacted with by the man. Meanwhile, the perceptual error introduced by the MLLMs still prevents the model from concluding the correct answer, causing the incorrect answer "five". Despite this incorrect answer, our framework still gives clear interpretable reasoning steps that identify the reason why the model is giving the wrong answer "five", and locates the root cause as the perceptual failure rather than incorrect reasoning. Such ability showcases that our LTR framework is able to increase the interpretability of existing MLLMs.

| Setting | | Causal-VidQA | NExT-QA |
|---|---|---|---|
| | | Acc@All | Acc@All |
| Baseline | | 68.0 | 80.4 |
| Full | | 70.2 | 82.7 |
| First Stage | w/o video | 68.9 | 81.5 |
| | w/o RAG | 69.3 | 82.2 |
| Second Stage | w/o A.V. | 69.7 | 82.0 |
| | w/o P.L.Q.A | 66.1 | 79.1 |
| | R. w/o video | 52.6 | 61.1 |

*Table 5.* Ablation studies. w/o video in first stage indicates that the video is not appear in first stage. w/o ans. ver. in second stage indicates that delete the In-process Answer Verification. w/o P.L.Q.A indicats that Perceptual Leaf Question Answering provides missing answers for leaf questions. R. w/o video indicates that no video is provided during the Video-aided Logical Reasoning.

## 6. Conclusion

In this work, we propose a novel two-stage Language-centric Tree Reasoning (LTR) framework that enhances the reasoning capabilities and transparency of MLLMs. In the first stage, LTR recursively generates a language-centric logical tree, utilizing language as the central driving force to gradually transform the complex congitive questions into simple perceptual qustions. In the second stage, aided by video content, LTR performs bottom-up logical reasoning within the tree to recursively derive the final answer with a complete, traceable reasoning path. To enhace the logical division ability of MLLMs, we exploit retrieval-augmented generation to guide question division. Extensive experiments across 11 VideoQA benchmarks demonstrate that the LTR framework significantly improves both accuracy and interpretability compared to state-of-the-art MLLMs. Overall, this work implements a traceable tree reasoning framework, paving the way for future research on language-centric video understanding from perception to cognition.

## Impact Statement

This paper presents work whose goal is to advance the field of Machine Learning. There are many potential societal consequences of our work, none which we feel must be specifically highlighted here.

## Acknowledgement

This work is supported by the National Natural Science Foundation of China (Grant No. 62402341, 62471287, 62372329), the Shanghai Municipal Science and Technology Major Project (Grant No. 2021SHZDZX0102), the Postdoctoral Fellowship Program of CPSF (GZC20241225) and National Key Research and Development Program of China (Grant No. 2024YFE0211000).

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

In this appendix, we provide additional experiments on various benchmarks in Section A to support the generalization ability and effectiveness of our framework. We also provide additional visual examples showcasing a more detailed analysis of how our LTR framework helps achieve successful video reasoning in **??**. Moreover, we provide discussions addressing potential common concerns in Section B. Finally, we conclude by discussing limitations and future work in Section C.

## A. Exprements on More Datasets

In this section, we conduct more experiments to demonstrate the effectiveness of our LTR framework in VideoQA. Specifically, we evaluate our framework under two settings: *i.e.*, open-ended and multiple-choice. The open-ended benchmarks include MSVD-QA (Xu et al., 2016), MSRVTT-QA (Xu et al., 2016), TGIF-QA (Jang et al., 2017), and ActivityNet-QA (Yu et al., 2019), which focus on relatively simple perceptual tasks like action and object recognition. The multiple-choice benchmarks comprise STAR (Wu et al., 2023), Ego-Schema (Mangalam et al., 2023), and Video-MME (Fu et al., 2024). Among them, STAR and Ego-Schema respectively target situated reasoning and long-video understanding, while Video-MME serves as an inference-only multifaceted VideoQA benchmark covering diverse tasks and video durations. We maintain identical evaluation protocols to our main experiments. For open-ended settings, GPT-3.5 assesses response quality using both accuracy and score metrics. For multiple-choice settings, MLLMs select answers from provided options based solely on questions and framework-generated responses.

### A.1. Experiments on Open-Ended Benchmarks

In this section, we evaluate the MSRVTT-QA, TGIF-QA, and ActivityNet-QA datasets where answers are short phrases, with results presented in Table 6. For TGIF-QA, we specifically assess the FrameQA, State Transition, and Repeating Action subsets. In the experimental results, we find that our model can improve the corresponding baseline MLLMs by 1%-2% in terms of accuracy. Although our LTR framework primarily enhances reasoning through *Language-centric Logical Trees*, these perceptual task gains remain consistent as existing MLLMs already possess strong visual perception capabilities that our method strategically utilizes.

| Method | MSVD-QA | | MSRVTT-QA | | TGIF-QA | | ActivityNet-QA | |
| --- | --- | --- | --- | --- | --- | --- | --- | --- |
| | Acc. | Score | Acc. | Score | Acc. | Score | Acc. | Score |
| Video-LLaVA (Zhang et al., 2023) | 51.6 | 2.5 | 59.2 | 3.5 | 70.0 | 4.0 | 45.3 | 3.3 |
| Video-ChatGPT (Maaz et al., 2023) | - | - | 49.3 | 2.8 | 51.4 | 3.0 | 35.2 | 2.7 |
| VideoLLaMA (Zhang et al., 2023) | - | - | 29.6 | 1.8 | - | - | 12.4 | 1.1 |
| LLaMA-VID (Li et al., 2024c) | 70.0 | 3.7 | 58.9 | 3.3 | - | - | 47.5 | 3.3 |
| LLaMA-Adapter (Zhang et al., 2024a) | 54.9 | 3.1 | 43.8 | 2.7 | - | - | 34.2 | 2.7 |
| Chat-UniVi (Jin et al., 2024a) | 65.0 | 3.6 | 54.6 | 3.1 | 60.3 | 3.4 | 45.8 | 3.2 |
| VideoChat (Li et al., 2023c) | 56.3 | 2.8 | 45.0 | 2.5 | 34.4 | 2.3 | - | 2.2 |
| Video-LaVIT (Jin et al., 2024b) | 73.2 | 3.9 | 59.3 | 3.3 | - | - | 50.1 | 3.3 |
| MiniGPT4-Video (Ataallah et al., 2024) | 73.9 | 4.1 | 58.8 | 3.3 | 72.2 | 4.1 | 45.9 | 3.4 |
| VideoChat2 (Li et al., 2024b) | 70.0 | 3.9 | 53.8 | 3.3 | 72.6 | 4.0 | 49.5 | 3.3 |
| + LTR | 71.6 | 3.9 | 55.6 | 3.4 | 73.9 | 4.0 | 51.1 | 3.3 |
| VideoLLaMA3 (Zhang et al., 2025) | 80.3 | 4.4 | 65.2 | 4.0 | 84.5 | 4.3 | 60.6 | 3.9 |
| + LTR | 82.0 | 4.5 | 67.1 | 4.1 | 86.1 | 4.4 | 62.4 | 4.0 |
| LLaVA-OneVision (Li et al., 2024a) | 78.8 | 4.2 | 63.4 | 3.6 | 81.0 | 4.1 | 56.0 | 3.5 |
| + LTR | 80.1 | 4.4 | 65.2 | 3.7 | 82.3 | 4.1 | 58.5 | 3.7 |
| Qwen2-VL (Wang et al., 2024) | 79.6 | 4.3 | 64.1 | 3.8 | 82.4 | 4.0 | 56.6 | 3.6 |
| + LTR | 81.0 | 4.4 | 65.8 | 3.8 | 84.6 | 4.0 | 58.4 | 3.6 |

*Table 6.* Experimental results on MSRVTT-QA, TGIF-QA, and ActivityNet-QA. The results in the white area are copied from the corresponding works, and the results in the blue area are reproduced by us using their published model weights and instructions. Acc. represents the accuracy.

### A.2. Experiments on Multiple-Choice Benchmarks

In this section, we evaluate our LTR framework on multiple-choice datasets including STAR, Ego-Schema and Video-MME. Experimental results are presented in Table 7 and Table 8. These benchmarks adopt the multiple-choice format due to historical limitations in generative capabilities of MLLMs, making it challenging to produce comprehensive and coherent

| Method | STAR | | | | | Ego-Schema |
|---|---|---|---|---|---|---|
| | Int. | Seq. | Pre. | Fea. | Avg. | |
| Video-LLaVA (Zhang et al., 2023) | 64.3 | 67.0 | 56.5 | 50.1 | 59.5 | 38.4 |
| LLaMA-VID (Li et al., 2024c) | - | - | - | - | - | 38.5 |
| VLAP (Wang et al., 2023) | 70.0 | 70.4 | 65.9 | 62.2 | 67.1 | - |
| Video-LaVIT (Jin et al., 2024b) | - | - | - | - | - | 37.3 |
| VideoChat (Li et al., 2023c) | 63.2 | 66.8 | 54.1 | 49.6 | 58.4 | - |
| VoT (Fei et al., 2024a) | 71.5 | 72.6 | 66.6 | 62.7 | 68.4 | - |
| VideoChat2 (Li et al., 2024b) | 54.1 | 64.9 | 68.2 | 64.4 | 62.9 | 53.3 |
| + LTR | 57.2 | 67.8 | 71.7 | 67.5 | 66.1 | 54.7 |
| VideoLLaMA3 (Zhang et al., 2025) | 60.1 | 68.8 | 68.7 | 62.6 | 65.0 | 56.7 |
| + LTR | 63.1 | 72.2 | 72.2 | 66.4 | 68.5 | 58.1 |
| LLaVA-OneVision (Li et al., 2024a) | 60.0 | 69.1 | 72.3 | 64.5 | 66.5 | 59.2 |
| + LTR | 62.8 | 72.4 | 75.2 | 67.9 | 69.6 | 61.0 |
| Qwen2-VL (Wang et al., 2024) | 65.2 | 74.9 | 70.5 | 68.7 | 69.8 | 65.0 |
| + LTR | 67.4 | 77.7 | 73.4 | 71.8 | 72.6 | 67.5 |

*Table 7.* Experimental results on STAR. "Int.", "Seq.", "Pre.", "Fea.", and "Avg." stand for "Interaction", "Sequence", "Prediction", "Feasibility", and "Average" respectively. The results in the white area are copied from the corresponding works or VoT (Fei et al., 2024a), and the results in the blue area are reproduced by us using their published model weights and instructions.

| Method | Short | Medium | Long | Average |
|---|---|---|---|---|
| Video-LLaVA (Zhang et al., 2023) | 45.3 | 38.0 | 36.2 | 39.9 |
| Chat-UniVi (Jin et al., 2024a) | 45.7 | 40.3 | 35.8 | 40.6 |
| SliME (Zhang et al., 2024b) | 53.3 | 42.7 | 39.8 | 45.3 |
| ShareGemini (Share, 2024) | 49.1 | 41.3 | 39.1 | 43.2 |
| VideoChat2 (Li et al., 2024b) | 50.0 | 39.3 | 40.3 | 43.2 |
| + LTR | 53.8 | 42.2 | 42.4 | 46.1 |
| VideoLLaMA3 (Zhang et al., 2025) | 76.6 | 62.8 | 54.6 | 64.6 |
| + LTR | 80.3 | 64.2 | 56.7 | 67.1 |
| LLaVA-OneVision (Li et al., 2024a) | 67.8 | 54.6 | 48.7 | 57.0 |
| + LTR | 71.7 | 57.2 | 50.0 | 59.6 |
| Qwen2-VL (Wang et al., 2024) | 69.8 | 58.6 | 51.8 | 60.0 |
| + LTR | 73.7 | 61.8 | 53.9 | 63.1 |

*Table 8.* Experimental results on Video-MME. The subtitle is not used during the evaluation. The results in the white area are copied from the corresponding works or Video-MME (Fu et al., 2024), and the results in the blue area are reproduced by us using their published model weights and instructions.

open-ended answers.

Overall, our LTR framework generally achieves 2%-4% improvements on these multiple-choice benchmarks, exceeding gains observed on open-ended benchmarks. This demonstrates particular effectiveness in reasoning-intensive tasks, aligning with our module designs: *Divide with Top-down Recursive Checking* and *Conquer with Bottom-up Tree Reasoning*. In the experimental results on STAR, our framework exhibits more significant improvements in temporal and causal reasoning tasks than that on descriptive tasks. This observation further demonstrates that our framework excels in complex cognitive tasks, enhancing the ability of existing MLLMs in temporal and causal reasoning. For Ego-Schema, LTR achieves modest gains (1.4%-2.5%) primarily due to its long-video reasoning focus versus 16-frame input limitation. This limitation constrains the capacity of MLLMs to perceive effective information from the video, causing more unreliable perceptual results in our Bottom-up Tree Reasoning stage, therefore resulting in a smaller performance boost. A similar trend is also evident in Video-MME, where the improvements on short and medium-length videos are more pronounced than that on long ones.

## B. Discussions

In this section, we discuss the rationale behind the designation of LTR framework, its inherent trade-offs, and its implications for multimodal reasoning. We first contextualize the balance between increased computational complexity and gains in interpretability enabled by hierarchical decomposition. Next, we explore the advantages of prioritizing linguistic logic over vision-centric approaches for structured reasoning. Finally, we analyze how training-free modular integration preserves the generalization power of foundation models while enabling task-specific adaptability.

## B.1. Complexity of Hierarchical Reasoning

The proposed two-stage LTR framework inherently introduces higher computational complexity compared to direct inference paradigms, yet this trade-off is strategically aligned with its advantages in test-time scalability and interpretability. By recursively decomposing questions into perceptual sub-problems and derive answers through bottom-up reasoning, the LTR framework mimics human-like hierarchical reasoning. While the tree-based structure prolongs inference time due to iterative decomposition and aggregation, it enables localized error analysis and modular optimization, reducing the risk of cascading failures in end-to-end systems. Crucially, the explicit tree generation provides transparent intermediate outputs (*e.g.*, sub-question perceptual checking and answer verification steps), allowing users to trace reasoning paths and diagnose failures. Though not ideal for latency-sensitive applications, this design ensures robustness in complex scenarios, as evidenced by consistent performance gains across diverse benchmarks. Future adaptations could incorporate adaptive depth control to dynamically balance precision and efficiency based on problem difficulty.

## B.2. Language-centric Logical Reasoning Paradigm

Unlike conventional VideoQA framework that prioritize visual feature extraction, our framework explicitly anchors reasoning in linguistic logic. The tree-generation stage first divides questions into logically coherent sub-questions, enforcing semantic rigor before engaging visual perception. This language-centric approach mitigates the inherent bias of vision-centric models, where dominant visual cues (*e.g.*, salient objects) may override abstract logical relationships. For instance, when resolving questions requiring multi-hop causality, the model establishes verifiable sub-goals through semantic decomposition rather than relying solely on visual correlations. While this paradigm sacrifices some efficiency in low-complexity perceptual tasks, it significantly enhances performance on benchmarks demanding structured reasoning, as logical constraints guide the visual search space. This aligns with cognitive principles where language scaffolds complex problem-solving, ensuring answers adhere to both visual evidence and contextual logic.

## B.3. Training-Free Generalization

By preserving pre-trained MLLMs as fixed modules and integrating Retrieval-Augmented Generation (RAG), our framework achieves task-specific adaptability without compromising generalization. Traditional fine-tuning methods often degrade foundational model capabilities when specializing for niche tasks, whereas our pipeline delegates logical structuring and visual perception to separate stages, avoiding parameter updates. The RAG component further enhances domain-specific reasoning by dynamically retrieving relevant knowledge (*e.g.*, the first-order sub-question struction for question division) during inference, eliminating the need for biased task-oriented training. Besides, the experimental results further underscore that systematic architectural innovation, rather than parameter optimization, can effectively balance specialization and generalization in multimodal reasoning systems.

# C. Limitation and Future Works

In our LTR framework, we introduce a language-centric logical reasoning approach comprising two key stages: *Divide with Top-down Recursive Checking* and *Conquer with Bottom-up Tree Reasoning* This framework generates and reasons over logical trees through language-structured representations. While demonstrating promising results, it exhibits limitations requiring further investigation and potential enhancements. Below we detail two primary constraints and propose future research directions.

## C.1. Detailed Understanding for Long Videos

Although we have observed that the accuracy on simple perceptual questions are generally higher than that of complex cognitive questions, it becomes difficult for the model to answer perceptual questions when facing long videos, since long videos may contain too much irrelevant information for answering simple perceptual questions. Such limitation is inherent in the baseline MLLMs due to their simple video-text alignment method based on frame-sampling.

In future work, improving the capability of MLLMs in understanding long videos is expected to enhance the performance of our LTR framework when facing complex long scenes, as such improvement would facilitate precise answering of perceptual questions and provide more accurate low-level information during the reasoning process.

Such limitation may be addressed through various directions. For example, a dense video representation (instead of frame-

sampling based representation) could introduce more detailed information that is lost during frame sampling. Moreover, a detailed spatio-temporal alignment from video to question may also help reduce noisy information for answering perceptual questions.

## C.2. System-2 Reasoning via Reinforcement Learning

Our LTR framework relies on the zero-shot capability of MLLMs to perform cognitive reasoning, which introduces two inherent constraints. First, while zero-shot methods avoid costly supervised training, they inherently depend on the pre-trained knowledge and reasoning biases of the base MLLMs, potentially limiting adaptability to domain-specific long-video scenarios. Second, the lack of explicit feedback mechanisms in zero-shot paradigms hinders iterative refinement of reasoning chains, a critical feature for handling complex spatio-temporal dependencies in long videos.

Recent advances in reasoning frameworks, such as reinforcement learning-based approaches (DeepSeek-AI et al., 2025) that enhance System-2 reasoning without requiring reward model training, suggest viable solutions. A promising direction involves integrating such methods into multimodal scenarios through our LTR framework. Furthermore, extending this learning paradigm to support curriculum learning strategies, progressing from short to long videos and from perceptual to cognitive tasks, could strengthen the hierarchical reasoning capabilities of MLLMs. This integration would not only improve accuracy in System-2 video reasoning but also enable more interpretable and auditable reasoning processes.

