# OpenReview forum: "Divide and Conquer: Exploring Language-centric Tree Reasoning for Video Question-Answering"
_ICML.cc/2025/Conference — ICML 2025 poster_

### Official Review · Reviewer_AFv7 · 2025-03-10

**Overall Recommendation:** 3

**Summary:**

The paper introduces Language-centric Tree Reasoning, a framework for VideoQA that hierarchically decomposes complex questions into a logical tree. It first recursively splits questions into perceptual sub-questions using linguistic cues and retrieval-augmented generation (RAG). Then, answers are aggregated bottom-up, guided by video content for verification. Experiments across multiple benchmarks show improved accuracy and interpretability over existing MLLMs.

## update after rebuttal

Thank the author for the rebuttal. I will keep my original rating, which was already positive.

**Claims And Evidence:**

LTR enhances reasoning accuracy and transparency in VideoQA by leveraging a hierarchical, language-driven decomposition. This claim is supported by:
- Quantitative gains of 1%–2% on open-ended tasks and 2%–4% on multiple-choice tasks.
- Improved compositional consistency metrics.
- Qualitative examples showing traceable reasoning steps that diagnose errors.

**Essential References Not Discussed:**

N/A

**Experimental Designs Or Analyses:**

- Dataset Coverage: Experiments conducted on a wide range of VideoQA datasets, including MSVD-QA, MSRVTT-QA, TGIF-QA, ActivityNet-QA, STAR, Ego-Schema, and Video-MME.
- Ablation Studies: Analysis of each component (e.g., RAG integration, video guidance, answer verification) to show their contributions.
- Qualitative Examples: Detailed reasoning paths for both successful cases and failure scenarios (highlighting where perceptual errors occur).

**Methods And Evaluation Criteria:**

Methods:
- Divide Stage: Recursively decompose a complex question into a language-centric logical tree using RAG to ensure semantic coherence.
- Conquer Stage: Perform video-aided bottom-up tree reasoning to aggregate sub-question answers, verify intermediate results, and derive the final answer.

Evaluation Criteria:
- Open-ended benchmarks assessed via GPT-3.5 (accuracy and scoring).
- Multiple-choice tasks evaluated by having MLLMs select from provided options.
- Detailed ablation studies and qualitative error analyses further validate each component.

**Other Comments Or Suggestions:**

N/A

**Other Strengths And Weaknesses:**

Weaknesses:
Heavy reliance on MLLMs’ zero-shot capabilities with no finetuning, which may limit adaptability in domain-specific tasks.

**Questions For Authors:**

Would incorporating fine-tuning for domain-specific scenarios help overcome the limitations of zero-shot reasoning in complex, long-video contexts?

**Relation To Broader Scientific Literature:**

- The work builds on previous VideoQA methods that utilize visual feature extraction and neural modular networks but distinguishes itself by focusing on interpretability.
- The authors compare against recent state-of-the-art models (e.g., VideoLLaMA, VideoChat2, Qwen2-VL) and position LTR as an approach that provides a clear reasoning trace.
- Aligns with literature on hierarchical reasoning and the cognitive basis for multi-hop question answering.

**Theoretical Claims:**

- Hierarchical Reasoning: Mimicking human System-2 reasoning by breaking down complex questions improves logical consistency.
- Language-Centric Approach: Anchoring the reasoning process in linguistic logic prior to engaging visual evidence reduces bias from overly salient visual cues.
- Training-Free Adaptability: Integrating pre-trained MLLMs with RAG allows for task-specific reasoning without additional fine-tuning, preserving generalization.

---

> ### Author Rebuttal · Authors · 2025-04-01
>
> Thank you for carefully reviewing our paper, acknowledging its strengths, and providing valuable suggestions for improvement
> If you have any further concerns, please feel free to raise them during the second-round rebuttal phase.
> As recommended by the official FAQ, we provide all figures and tables via the [anonymous link](https://anonymous.4open.science/r/ICML25-3189-7D31/README.md).
>
> ### Domain Adaptability and Finetuning on Domain Specific Tasks
> In general, finetuning MLLMs in a specific domain significantly enhances performance by providing a more solid basis for reasoning. We further detail finetuning for different modules as follows:
> * Divide with Top-down Recursive Checking: Finetuning is generally unnecessary for this stage since question decomposition primarily relies on language understanding, with visual content serving as a supplementary element where coarse understanding is sufficient for most scenarios. However, when encountering different question distributions or videos with distinct characteristics (e.g., 360-degree videos or extremely long videos), finetuning may offer significant benefits.
> * Perceptual Leaf Question Answering: This step can be readily finetuned using any VideoQA dataset, which would improve the accuracy of leaf question answers and, consequently, enhance subsequent reasoning stages.
> * Video-Aided Logical Reasoning and In-Process Answer Verification: Although finetuning these steps can benefit the framework, it requires additional effort to construct corresponding reasoning data for the specific domain.
>
> To further validate the effectiveness of finetuning at different stages, we conducted experiments on AGQA-Decomp by finetuning our entire framework, leveraging the compositional graph and detailed answers provided in AGQA-Decomp. In [Table1](http://anonymous.4open.science/r/ICML25-3189-7D31/R-AFv7/Table1.png), we compare the performance of Qwen2-VL and VideoChat2 on AGQA-Decomp with and without finetuning. For the main question, our LTR framework improves the baseline MLLMs by 3%–4%, and the finetuning strategy further boosts LTR by an additional 3%–4%, leading to a total improvement of 7%–8% in accuracy. For sub-questions, the improvement is even larger, 12%–13%, because the Perceptual Leaf Question step is more amenable to enhancement in a specific setting. Moreover, the improvement in terms of $c$-$F_1$ is approximately 15%–16%, largely attributable to the increased sub-question accuracy and the hierarchical information aggregation and logical inference in Video-Aided Logical Reasoning. We thank the reviewer for this suggestion and will incorporate these experimental results and discussions in the following version of the paper.
>
> We thank the reviewer again for the detailed reviewing, and hope our rebuttal have solved the proposed concerns and would strengthen the reviewer's confidence in judging positively for this paper.

---

### Official Review · Reviewer_DfhN · 2025-03-14

**Overall Recommendation:** 3

**Summary:**

This paper proposes Language-centric Tree Reasoning (LTR), a training-free, model-agnostic framework that enhances reasoning capabilities and interpretability in Video Question Answering (VideoQA) by using MLLMs. LTR addresses the limitations of existing MLLMs, such as opacity and lack of controllability in their reasoning processes. The framework operates by recursively generating a language-centric logical tree based on the input question and incorporating video content to create leaf nodes. These leaf nodes represent simple perceptual questions that can be answered by MLLMs. LTR then performs bottom-up reasoning through the tree, leveraging MLLM responses to the leaf node questions and verifying consistency with visual evidence. This process culminates in an answer to the original question and a traceable reasoning path. Experiments on 11 VideoQA benchmarks using four different MLLMs demonstrate that LTR improves reasoning accuracy and provides a more transparent and verifiable VideoQA system. Ablation studies analyze the effectiveness of individual components within the framework, and case studies showcase its enhanced error tolerance and explainability.

**Claims And Evidence:**

The evidence presented largely supports the claims made in the submission, offering a convincing case for the effectiveness of the LTR framework.

**Essential References Not Discussed:**

The paper provides a comprehensive overview of related work.

**Experimental Designs Or Analyses:**

The experimental design appears generally sound, encompassing a suitable range of evaluations. Testing the LTR framework with four different open-source MLLMs across 11 benchmarks, covering both open-ended and multiple-choice question types, provides a comprehensive assessment of its effectiveness. The chosen evaluation metrics are appropriate for VideoQA tasks. Furthermore, including a thorough ablation study allows for a detailed analysis of the contributions of individual components within the LTR framework. However, certain specific questions regarding the experimental setup and results are raised in the "Other Strengths and Weaknesses" section.

**Methods And Evaluation Criteria:**

The proposed methods for enhancing VideoQA through language-centric tree reasoning are reasonable to the problem.

**Other Comments Or Suggestions:**

- On page 2, line 83, left column, "oof" appears to be a typographical error and should be corrected to "of."
  - In section 3.2.2, "Video-Aided Logical Reasoning," the provided example refers to "Figure 2 (red box)." However, the explanation seems to correspond to Figure 1, not Figure 2. The authors should verify the figure reference and correct it accordingly for clarity and accuracy.

**Other Strengths And Weaknesses:**

Strengths:
  - The language-centric tree reasoning framework appears novel and offers a promising approach to improving interpretability and controllability in VideoQA.
  - The proposed framework's training-free and model-agnostic nature enhances its practicality and broad applicability.


Weaknesses:
  - The LTR framework, as described, involves multiple steps and inferences to process a given question and generate the reasoning tree. While this is acceptable for research purposes, practical applications require a fully automated pipeline integrated with the model, eliminating manual intervention. The paper needs to address the feasibility of such an automated pipeline and discuss how the process of generating the tree can be seamlessly integrated into a real-world VideoQA system. The lack of a clear automation strategy limits the scalability and generalizability of the proposed approach.
  - The current description of the LTR framework suggests its applicability might be limited to specific question types. It remains unclear how the framework would handle more complex or nuanced questions, such as those requiring summarization (e.g., "Summarize what happened in this video") or those involving negation in multiple-choice scenarios (e.g., "Which of the following statements is false?"). The authors should clearly define the types and scope of questions that the LTR framework can handle and explicitly address these limitations. A discussion of the limitations regarding question types should also be included in the limitations section.
  - While the reported experimental results show improvements across various metrics, the paper would benefit significantly from richer qualitative analysis. Including more detailed case studies would provide valuable insights into the LTR framework's practical impact. Specifically, the authors should provide examples of:
    - Questions correctly answered with LTR that were previously answered incorrectly by the baseline MLLMs.
    - Questions that remain unanswered even with LTR.
    - Questions answered correctly by the baseline MLLMs but incorrectly answered after applying LTR.

    These case studies would provide the community with a deeper understanding of the LTR framework's strengths and weaknesses, facilitating further research and development in this important area.

**Questions For Authors:**

- The proposed LTR framework presents a compelling approach to incorporating reasoning into VideoQA. However, the authors should elaborate on how the process, particularly the In-Process Answer Verification stage, ensures the correctness of intermediate responses within the reasoning tree. What mechanisms are in place to prevent and handle potential errors during this stage, such as incorrect verification by the model itself? How would the framework perform if an incorrect answer is generated at an intermediate node? Could this lead to a cascading effect, ultimately resulting in an incorrect final answer?
  - The experiments utilize open-source models. Have the authors explored applying the LTR framework to any closed-source models?

**Relation To Broader Scientific Literature:**

This work addresses the limitations of current MLLMs in VideoQA, particularly their lack of transparent System-2 reasoning. It proposes a novel language-centric reasoning framework, offering a potential solution to the interpretability challenges faced by existing approaches like VoT and DSTN.

**Theoretical Claims:**

This submission does not present any theoretical claims requiring formal proofs. The focus is on the empirical evaluation of the proposed framework.

---

> ### Author Rebuttal · Authors · 2025-04-01
>
> Thank you for carefully reviewing our paper, acknowledging its strengths, and providing valuable suggestions for improvement
> If you have any further concerns, please feel free to raise them in the rebuttal comment.
> As recommended by the official FAQ, we provide all figures via the [anonymous link](https://anonymous.4open.science/r/ICML25-3189-7D31/README.md).
>
> ### Automated Pipeline
> We agree that automation is critical for practical applications. Our two-stage approach, first generating the language-centric logical tree, then performing bottom-up reasoning through tree validation, handles VideoQA automatically.
> For each question, the system recursively decomposes it into simpler sub-questions until all leaf nodes are perceptive, using retrieved few-shot examples as guidance.
> The resulting tree is processed bottom-up to analyze both the video and question, yielding interpretable answers.
> Furthermore, while LTR is designed for full automation, it also allows for human intervention in sensitive scenarios, enabling users to adjust intermediate responses for better accuracy.
>
> ### Applicable Question Types
> We provide two figures to illustrate how LTR framework handles summarization and negation in multiple-choice scenarios. For the summarization question([Figure1](https://anonymous.4open.science/r/ICML25-3189-7D31/R-DfhN/Figure1.png)), LTR naturally decomposes the video into parts and generates a Language-centric Logical Tree that prompts a summary for each segment, leading to a comprehensive summary of the main question. In the case of negation in multiple-choice questions([Figure2](https://anonymous.4open.science/r/ICML25-3189-7D31/R-DfhN/Figure2.png)), the framework breaks down the question by checking each statement individually and then integrates the results to determine which statement is false, closely mirroring human reasoning.
>
> However, for questions that are linguistically simple yet require complex cognitive visual reasoning, such as “Is there a thief?”. LTR may struggle to generate a reasonable Language-centric Logical Tree. In these cases, the question is often misclassified as perceptual due to its simplicity, even though answering it requires detailed motion analysis and a comprehensive understanding of the video content.
> We will include these discussions in the following version.
>
> ### More Qualitative Results
> We further provide three cases to illustrate LTR’s performance in different settings.
> In [Figure3](https://anonymous.4open.science/r/ICML25-3189-7D31/R-DfhN/Figure3.png), for “Questions unanswered with LTR,” the counting sub-question misidentify background objects due to perceptual miscounting and a logic trap, resulting in an incorrect final answer despite a transparent reasoning process.
> In [Figure4](https://anonymous.4open.science/r/ICML25-3189-7D31/R-DfhN/Figure4.png), for “Questions correctly answered with LTR and incorrectly answered by baseline,” LTR accurately deduced the stationary object’s properties by intersecting responses from multiple sub-questions and leveraging visual-aided reasoning, whereas the baseline relied solely on perceptual cues.
> In [Figure5](https://anonymous.4open.science/r/ICML25-3189-7D31/R-DfhN/Figure5.png), for “Questions correctly answered by baseline and incorrectly answered with LTR,” the baseline produce a correct answer without any explanation. However, an erroneous leaf response in LTR led to an incorrect final answer, even though the intermediate video content analysis remained robust.
>
> ### Errors in Intermediate Nodes
> Errors or hallucinations in reasoning are inevitable, but prior study [1] show that providing more context or sampling multiple answers can help suppress them. In LTR, detailed sub-question decomposition offers comprehensive context that reduces hallucinations and minimizes cascading errors, as evidenced by improved sub-question accuracy. So, an error at an intermediate node is less likely to cascade and adversely affect the final answer.
>
> The In-Process Answer Verification stage further mitigates errors by prompting the model to check both reasoning logic and consistency between answers and visual content.
> Although no system can fully eliminate errors, this verification step increases the reliability of intermediate responses, as confirmed by experiments in Table5 of the submission.
> We will include these discussions in the following version.
>
> [1] Fei, H. et.al. Video-of-thought: Step-by-step video reasoning from perception to cognition. In ICML, 2024
>
> ### Performances on Closed-Source Models.
> To valid the effectiveness of LTR on closed source models, we conduct experiments using GPT-4o on EgoSchema and MVBench, with 16 uniformly sampled frames per video.
> As summarized in [Table1](https://anonymous.4open.science/r/ICML25-3189-7D31/R-DfhN/Table1.png), LTR yield significant accuracy improvements on these benchmarks, demonstrating its generalizability across various models.
>
> ### Minor Issue
> All typos will be revised in the following version.

---

### Official Review · Reviewer_8P4e · 2025-03-14

**Overall Recommendation:** 3

**Summary:**

This paper introduces Language-centric Tree Reasoning (LTR), a framework to enhance the reasoning of MLLMs. It uses MLLMs to first hierarchically break down a question into sub-questions, then conquer the question by answering the sub-questions in a bottom-up way. In the experiments, LTR is applied to four state-of-the-art MLLMs and it can consistently improve their performance on 11 video question answering benchmarks.

**Claims And Evidence:**

1. The paper claims that the proposed LTR can improve the reasoning capability of MLLMs. The consistent improvements on four MLLMs across 11 benchmarks support this claim.
2. The paper claims that the design choices in the LTR are essential to the overall performance. The ablation studies in Section 4.5 support this claim.

**Essential References Not Discussed:**

N/A

**Experimental Designs Or Analyses:**

To demonstrate the effectiveness of the LTR framework, it is applied on a few pre-trained MLLMs and it is shown that the performance can be consistently improved. However, the experiments do not compare LTR with other baseline methods that can improve the reasoning capability of MLLMs. The proposed LTR can be proved effective only when it can outperform the baselines. A few simple baselines include:
1. Chain-of-Thought prompting [1], which prompts the models to "think step-by-step".
2. LTR needs to execute an MLLM for multiple times to answer one question. Therefore, it is a test-time scaling approach. The following test-time scaling baselines should be considered:

(a) Majority Voting [2], which uses the MLLM to sample N answers and selects the most frequent one as the response.

(b) Best-of-N [2], which samples N answers and uses a reward model to score the answers. The one with the highest score is selected as the response. The reward model can be the evaluated MLLM itself.

Especially, the N used in the baselines above should be comparable to the average number of MLLM executions in LTR.

[1] Chain-of-Thought Prompting Elicits Reasoning in Large Language Models. Jason Wei et al. NeurIPS 2022.

[2] Let's Verify Step by Step. Hunter Lightman et al. ICLR 2024.

**Methods And Evaluation Criteria:**

The proposed method is well-motivated.

**Other Comments Or Suggestions:**

**Typos:**
1. L82-L83: off -> of
2. L380-L381: existace -> existence

**Writing:**
1. In L46-L47, the concept of "System-2 reasoning" should be explained.
2. In Section 4.5, qualitative examples are highly appreciated to demonstrate how each module affects the reasoning of the model.
3. In Section 4.6, the base model used in the qualitative example should be mentioned.

**Rating justification:** The proposed method is well-motivated and can effectively improve the performance of MLLMs. However, due to the lack of comparison with any existing baseline that can improve MLLM reasoning, I cannot vote for the acceptance of this paper. I am open to adjust my rating after rebuttal.

**Other Strengths And Weaknesses:**

**Other Strengths:**
1. The proposed LTR framework is training-free and can be directly employed on pre-trained MLLMs.
2. Multiple up-to-date MLLMs are evaluated in the experiments.
3. The paper writing is clear and easy to follow.

**Other Weaknesses:**
1. The question of the qualitative example in Figure 3 is too easy. The answer can be easily guessed even without the video provided. To demonstrate the effectiveness of the proposed method, a more challenging question should be used as a qualitative example.

**Questions For Authors:**

N/A

**Relation To Broader Scientific Literature:**

The previous MLLM reasoning works do not capture the logical structure of questions. The LTR framework proposed in this work leverages the logical structure and improves the MLLM performance on video question answering benchmarks.

**Theoretical Claims:**

This paper does not involve theoretical claims.

---

> ### Author Rebuttal · Authors · 2025-04-01
>
> Thank you for carefully reviewing our paper, acknowledging its strengths, and providing valuable suggestions for improvement
> If you have any further concerns, please feel free to raise them during the second-round rebuttal phase.
> As recommended by the official FAQ, we provide all figures and tabels via the [anonymous link](https://anonymous.4open.science/r/ICML25-3189-7D31/README.md).
>
> ### Comparison with More Baseline Methods.
> To further validate the effectiveness of our LTR, we compare the it with three baselines (CoT, Major Voting, and Best of N), across AGQA-Decomp, MVBench, and CausalVidQA using VideoChat2 as the MLLM, as shown in [Table1](https://anonymous.4open.science/r/ICML25-3189-7D31/R-8P4e/Table1.png).
> For fair comparison, the repetition count (N) for all baselines is set to match the number of MLLM executions in LTR for various questions.
> For CoT, we carefully construct [five text-based examples](https://anonymous.4open.science/r/ICML25-3189-7D31/R-8P4e/Figure3.png) as prompts generated by GPT-4o.
> For Best of N, VideoChat2 also serves as the reward model.
> These comparisons will be included in the following version.
>
>
> In [Table1](#table1), although CoT, Major Voting, and Best of N improve model accuracy on three datasets, their gains remain relatively moderate compared with our LTR, largely due to the lack of detailed information in these methods.
> Specifically, the Major Voting and Best of N do not provide detailed sub-question analysis and the CoT offers only simple reasoning examples that constrain the reasoning ability of MLLMs, however, our LTR delivers a structured, comprehensive context and a detailed reasoning process that clearly demonstrates how the answer is deduced, thereby enhancing both accuracy and interpretability.
>
> Furthermore, the compared methods contribute little to the compositional consistency of models, as they lack intra-question information exchange. In contrast, our LTR employs a hierarchical information aggregation and logical inference procedure that yields a consistent and accurate reasoning process, ultimately achieving superior performance in accuracy and compositional consistency on several VideoQA benchmarks (***i.e.***, AGQA-Decomp, MVBench, and CausalVidQA) while also offering enhanced interpretability and controllability.
>
> ### More Complex Qualitative Results and Module Analysis
> To validate the effectiveness of our LTR framework, we provide two complex qualitative examples in [Figure 1](https://anonymous.4open.science/r/ICML25-3189-7D31/R-8P4e/Figure1.png) and [Figure 2](https://anonymous.4open.science/r/ICML25-3189-7D31/R-8P4e/Figure2.png) in our response, along with a detailed analysis for the first example.
>
> In [Figure1](https://anonymous.4open.science/r/ICML25-3189-7D31/R-8P4e/Figure1.png), the video depicts a scenario where a man is struck in the crotch by a dog, causing him to bend over and hold his crotch. In Stage 1, the Divide with Top-down Recursive Checking successfully decomposes the main question into a language-centric logic tree with simpler sub-questions that precisely target perceptual aspects (e.g., detecting the man, the dog, and the actions performed). In Stage 2, the Video-Aided Logical Reasoning module integrates both logical cues and visual evidence to infer answers for each non-leaf node. Crucially, the In Process Answer Verification stage is applied to ensure consistency across intermediate responses; for instance, it corrects the answer for non-leaf question [inter_2] by cross-validating the reasoning with the visual content.
>
> Furthermore, we also update more qualitative results along with comprehensive discussion, covering both successful and failure cases, to better illustrate the impact of each module within our LTR framework.
> The revised figures and detailed descriptions are available in our response to Reviewer DfhN.
> We will include these examples and discussions in the following version.
>
> ### Explanation of System-2 Reasoning
> The term system-2 originates from psychology and cognitive science, where the dual process theory delineates human reasoning into two distinct processes: system-1, which is fast, intuitive, situational, and perceptual, and system-2, which is slow, logical, abstract, and cognitive [1].
> Notably, CoT, Major Voting, and Best of N are also typical examples of system-2 reasoning models.
> We will explain more on system-2 reasoning and its relevance to computer science in the follow version.
>
> ### Minor Issue
> The MLLM employed in the qualitative analysis is VideoChat, and all the typos will be corrected in the following version.
>
> [1] Evans, J. S. In two minds: dual-process accounts of reasoning. Trends in Cognitive Sciences, 7(10):454–459, 2003. ISSN
> 1364-6613.

---

> > ### Comment · Reviewer_8P4e · 2025-04-05
> >
> > The authors' responses addressed my concerns. I highly appreciate the comparison with a few baseline methods. I have raised my rating to weak accept.

---

### Decision · Program_Chairs · 2025-05-01

**Decision:**

Accept (poster)

**Comment:**

After the rebuttal, two reviewers have increased their ratings, and all reviewers provide positive ratings (3 Weak Accepts). The mentioned concerns have been addressed. On the other hand, there is no reviewers who "strongly" support this submission, and one reviewer is not an expert in this area. Thus, we recommend **Weak Accept** for this submission if there is room in the program.